# Proteomic Profiling of HDL in Newly Diagnosed Breast Cancer Based on Tumor Molecular Classification and Clinical Stage of Disease

**DOI:** 10.3390/cells13161327

**Published:** 2024-08-09

**Authors:** Monique de Fatima Mello Santana, Maria Isabela Bloise Alves Caldas Sawada, Douglas Ricardo Souza Junior, Marcia Benacchio Giacaglia, Mozania Reis, Jacira Xavier, Maria Lucia Côrrea-Giannella, Francisco Garcia Soriano, Luiz Henrique Gebrim, Graziella Eliza Ronsein, Marisa Passarelli

**Affiliations:** 1Laboratório de Lípides (LIM10), Hospital das Clínicas (HCFMUSP), Faculdade de Medicina, Universidade de Sao Paulo, Sao Paulo 01246-000, Brazil; 2Programa de Pós-Graduação em Medicina, Universidade Nove de Julho, Sao Paulo 01525-000, Brazil; isabelacaldas028@gmail.com (M.I.B.A.C.S.); moszania@gmail.com (M.R.);; 3Grupo de Saúde de Curitiba (GSAU-CT), CINDACTA II, Brazilian Air Force, Curitiba 82510-901, Brazil; 4Laboratório de Proteômica Aplicada à Processos Inflamatórios, Instituto de Química, Universidade de Sao Paulo, Sao Paulo 05508-900, Brazilronsein@iq.usp.br (G.E.R.); 5Unidade Básica de Saúde Dra. Ilza Weltman Hutzler, Sao Paulo 02472-180, Brazil; 6Laboratório de Carboidratos e Radioimunoensaio (LIM18), Hospital das Clínicas (HCFMUSP), Faculdade de Medicina, Universidade de Sao Paulo, Sao Paulo 01246-000, Brazil; maria.giannella@fm.usp.br; 7Laboratório de Emergências Clínicas (LIM51), Hospital das Clínicas (HCFMUSP), Faculdade de Medicina, Universidade de Sao Paulo, Sao Paulo 01246-000, Brazil; 8Centro de Referência da Saúde, Mulher-Hospital Pérola Byington, Sao Paulo 01215-000, Brazil

**Keywords:** HDL, proteomics, breast cancer, triple-negative breast cancer

## Abstract

The association between high-density lipoprotein (HDL) cholesterol and breast cancer (BC) remains controversial due to the high complexity of the HDL particle and its functionality. The HDL proteome was determined in newly diagnosed BC classified according to the molecular type [luminal A or B (LA or LB), HER2, and triple-negative (TN)] and clinical stage of the disease. Women (n = 141) aged between 18 and 80 years with BC, treatment-naïve, and healthy women [n = 103; control group (CT)], matched by age and body mass index, were included. Data-independent acquisition (DIA) proteomics was performed in isolated HDL (D = 1.063–1.21 g/mL). Results: Paraoxonase1, carnosine dipeptidase1, immunoglobulin mMu heavy chain constant region (IGHM), apoA-4, and transthyretin were reduced, and serum amyloid A2 and tetranectin were higher in BC compared to CT. In TNBC, apoA-1, apoA-2, apoC-2, and apoC-4 were reduced compared to LA, LB, and HER2, and apoA-4 compared to LA and HER2. ComplementC3, lambda immunoglobulin2/3, serpin3, IGHM, complement9, alpha2 lysine rich-glycoprotein1, and complement4B were higher in TNBC in comparison to all other types; complement factor B and vitamin D-binding protein were in contrast to LA and HER2, and plasminogen compared to LA and LB. In grouped stages III + IV, tetranectin and alpha2-macroglobulin were reduced, and haptoglobin-related protein; lecithin cholesterol acyltransferase, serum amyloid A1, and IGHM were increased compared to stages I + II. Conclusions: A differential proteomic profile of HDL in BC based on tumor molecular classification and the clinical stage of the disease may contribute to a better understanding of the association of HDL with BC pathophysiology, treatment, and outcomes.

## 1. Introduction

Breast cancer (BC) is the most commonly occurring cancer overall and stands out as the predominant cancer affecting women. This complex disease is marked by its heterogeneous nature, and encompasses distinct biological and histological traits, clinical manifestations, and responses to therapy. The molecular classification of BC plays a crucial role in facilitating precise diagnosis, selecting appropriate treatments, and accurately predicting prognosis, especially in the context of hormonal and anti-HER2-targeted therapies [1].

The luminal A (LA) subtype, constituting 40% to 50% of invasive BC cases, is characterized by the expression of estrogen (ER) and progesterone (PR) receptors, correlating with a more favorable prognosis. Conversely, luminal B (LB) displays a lower expression of ER-related genes, variable HER2-related gene expression, and an increased expression of proliferation-related genes compared to LA. The HER2-overexpressing subtype, comprising approximately 15% of all invasive BC cases, exhibits a heightened expression of HER2/HER2 signaling-associated genes, leading to a more aggressive clinical course, albeit with a favorable response to anti-HER2-targeted therapy. Triple-negative breast cancer (TNBC), representing 10–15% of total BC cases, lacks ER and PR receptors as well as HER2 expression. TNBC is characterized by its high metastatic potential, heterogeneity, and an overall poor prognosis. The contribution of plasma lipids and lipoproteins to the development and prognosis of BC has been addressed in many ways [2]. Recent data have shed light on a distinct lipid profile in TNBC, indicating elevated plasma levels of triglycerides (TG), total cholesterol (TC), non-HDL cholesterol (nonHDLc), and apolipoprotein B (apo B), suggesting a potential contribution to lipid channeling into the tumor microenvironment [3].

Although classically related to atheroprotection, the role of high-density lipoprotein cholesterol (HDLc) has gained substantial attention in understanding the development and progression of BC. While HDLc is commonly inversely linked to BC incidence and unfavorable disease outcomes, instances of no association and even positive correlations have been demonstrated [4,5,6,7,8,9]. The controversies seem to lie in the fact that the determination of HDLc may not be the best metric for inferring the functional properties of HDL.

HDL mediates cell cholesterol removal, limiting the sterol content necessary for tumor replication and metastasis. Moreover, it counteracts oxidative and inflammatory stress that aggravates tumor biology. The complexity of the HDL structure is determined by its composition in bioactive lipids, proteins, and microRNAs (miRs), and dictates its functionality [10]. The analysis of the HDL proteome enabled the identification of dozens of proteins associated with various HDL subfractions, modulating different functional properties of these lipoproteins [11]. Additionally, HDL proteomics can support the identification of biomarkers related to disease and therapeutic responses.

Considering the heterogeneity of BC, functional phenotypic aspects have been determined based on protein expression profiles [12,13,14]. The HDL proteome has been studied under various conditions to understand its role in chronic diseases, especially atherosclerosis. However, no data currently exist regarding HDL’s role as a carrier particle for proteins that might modulate BC, helping to provide new insights into potential markers of disease progression, overall survival, and treatment goals. HDL proteomics were analyzed in newly diagnosed BC cases and compared to matched CT women. A different profile of protein expression was observed in BC cases, varying according to the molecular type and clinical stage of the disease.

## 2. Materials and Methods

### 2.1. Population of the Study

The study comprised 143 cases of BC selected from a larger cohort of 201 recently diagnosed women who had not undergone treatment, aged between 18 and 80 years, encompassing all clinical stages and with the documented molecular classification of the tumor. These participants were recruited from Hospital Pérola Byington, Sao Paulo, Brazil. Furthermore, 141 women were chosen from a cohort of 157 individuals without any prior cancer history, recruited from Universidade de São Paulo and Unidade Básica de Saúde Dra. Ilza Weltman Hutzler, constituting the control group (CT). Exclusion criteria included diabetes mellitus, chronic kidney disease (estimated glomerular filtration rate < 60 mL/min/1.73 m^2^), autoimmune diseases, smoking, alcoholism, the use of contraceptives, hormone replacement therapy, pregnancy, a previous history of any cancer, and in situ breast disease. All participants were informed about the study, and their informed written consent, previously approved by institutional Ethics Committees following the Declaration of Helsinki, including approval for publication (Universidade Nove de Julho, #3.139.460; Centro de Referência da Saúde da Mulher, Hospital Pérola Byington, #3.225.220; and Hospital das Clínicas da Faculdade de Medicina da Universidade de São Paulo, #3.317.909), was obtained. The molecular classification of tumors was obtained from medical records through immunohistochemical analysis, according to the guidelines of the American College of Pathologists [15]. Samples positive for ER and PR were identified when more than 1% of tumor cells exhibited positive nuclear staining of moderate to strong intensity on immunohistochemistry. They were then classified as LA and LB based on a Ki67 index below or above 14%. Samples showing more than 10% of invasive tumor cells with strong HER2 staining in the plasma membrane were deemed HER2 positive. In cases of moderate staining in more than 10% of the cells or strong staining in less than 10% of the cells, the sample underwent re-evaluation through in situ hybridization. It was considered positive if a HER2/centromere ratio exceeded 2.0 or a HER2/centromere ratio was below 2.0 with a mean HER2 surpassing 6 signals per cell (more than 120 signals in 20 nuclei). Tumor samples lacking the expression of hormonal or HER2 receptors were classified as TNBC [15]. Subjects were categorized to the clinical stage of disease as stages I, II, III, and IV, according to the TNM, 8th Edition [16].

### 2.2. Blood Collection and Determination of Plasma Lipid Profile

Venous blood was collected after 12 h of fasting and plasma was immediately isolated after centrifugation (3000 rpm, 4 °C, 15 min). Plasma lipid concentrations were determined by enzymatic techniques (Roche Diagnostics, Sao Paulo, Brazil), low-density lipoprotein cholesterol (LDLc) was calculated by the Friedewald formula [16], and VLDLc as TG/5. ApoB was quantified by immunoturbidimetry (Randox Lab. Ltd., Crumlin, UK).

### 2.3. HDL Isolation

High-density lipoprotein (HDL; D = 1.063–1.21 g/mL) was isolated from plasma by discontinuous density ultracentrifugation for 24 h, at 4 °C, 100,000× *g*, (SW40 rotor; L80-Beckman ultracentrifuge Palo Alto, CA, USA). Lipoproteins were carefully removed by a vacuum and HDL fraction was stored at −80 °C in a 5% sucrose solution. HDL composition in apoA-1 was determined by immunoturbidimetry (Randox Lab. Ltd., Crumlin, UK) and in lipids (TC, TG, and PL) by enzymatic techniques (Roche Diagnostics).

### 2.4. HDL Digestion for Proteomics

The Bradford assay (Bio-Rad, Hercules, CA, USA) determined total protein concentration in isolated HDL. Five micrograms of HDL protein were diluted in 50 mM of an ammonium bicarbonate buffer (Sigma-Aldrich, St. Louis, MO, USA) containing 0.01% ProteaseMAX MS-compatible surfactant (Promega). Proteins were reduced with 5 mM of dithiothreitol (Bio-Rad), for 1 h at 37 °C, alkylated with 15 mM of iodoacetamide (Bio-Rad) for 30 min and excess iodoacetamide was quenched using 2.5 mM of dithiothreitol for 15 min at room temperature. Proteins were then digested with trypsin [1:40, *w*:*w*, trypsin (Promega, Madison, WI, USA): HDL proteins] for 4 h at 37 °C. A second aliquot of trypsin (1:50, *w*:*w*) was added to the samples [17,18]. After overnight incubation at 37 °C, HDL peptide samples were acidified using 0.5% trifluoroacetic acid (Sigma-Aldrich) and desalted using the C18-StageTip protocol [19]. All steps were conducted in a single batch to eliminate inter-assay variability. Before MS analysis, samples were resuspended in 0.1% formic acid (Fisher Chemical, Zurich, Switzerland) at a concentration of 50 ng/µL. One microliter (50 ng) of each sample was injected into the LC-MS/MS system. MS proteomics data have been deposited to the Mass Spectrometry Interactive Virtual Environment (MassIVE) with access via ftp://massive.ucsd.edu/v08/MSV000095160/ and http://doi.org/10.25345/C5J38KV56. (accessed on 7 August 2020).

### 2.5. Liquid Chromatography–Mass Spectrometry (LC-MS/MS) Analyses

Digested HDL proteins were loaded onto a trap column (nanoViper C18, 3 μm, 75 μm × 2 cm, Thermo Scientific, Waltham, MA, USA) and eluted onto a C18 column (nanoViper, 2 μm, 75 μm × 15 cm, Thermo Scientific). Peptide analyses were performed employing an Easy-nLC 1200 UHPLC system coupled to an Orbitrap Fusion Lumos tribrid mass spectrometer equipped with a nanospray FlexNG ion source (Thermo Scientific, 2150 V), in a 39 min gradient and normalized collision energy of 30 for HCD fragmentation. The LC system was initially set at a flow rate of 300 nL/min using pumps A (0.1% formic acid) and B (80% acetonitrile in 0.1% formic acid). A linear gradient from 5 to 28% B was achieved in 25 min, followed by another linear gradient from 28 to 40% B in 3 min. Solvent B was increased to 95% B in one minute to wash the system, which was maintained for another 10 min (350 nL/min), before re-equilibrating the system for another run.

Data-dependent acquisition (DDA) was used to build a spectral library. With this purpose, DDA precursor ions were identified using an MS1 resolution of 120,000 (at *m*/*z* 200) with an AGC target set to 5 × 10^5^, a maximum injection time of 50 ms, and a full scan range of 350–1550 *m*/*z*. Fragment ions were analyzed in MS2 mode with a resolution of 30,000 (at *m*/*z* 200), with a standard AGC target and maximum injection time of 54 ms. For data-independent acquisition (DIA), transitions were monitored in centroid mode with a resolution of 30,000 (at *m*/*z* 200), an AGC target of 5 × 10^5^, a precursor isolation range between 400 and 1000 *m*/*z*, a scan range of product ions between 150 and 1650 *m*/*z*, a maximum injection time of 54 ms and staggered isolation windows of 24 *m*/*z* with 0.5 *m*/*z* margins. A precursor ion MS1 full scan was also acquired in profile mode between each cycle using 30,000 resolution and *m*/*z* range between 350 and 1550. All MS analyses were performed using orbitrap as the mass analyzer and RunStart EASY-IC as internal calibration.

### 2.6. Sample Randomization and Quality Controls (QCs)

DDA proteomics were used to acquire data to build a spectral library to quantify HDL proteins by DIA [18,20,21]. Thus, for DDA, 15 pools were constructed, consisting of 20 randomly selected samples.

For DIA quantifications, samples from both experimental groups were randomly injected into the MS. Twenty samples were randomly chosen to compose a pooled sample used as a quality control. This QC pooled sample was digested in 12 replicates to account for digestion variability (Digestion QC). Furthermore, a pooled QC sample was injected 19 times, intercalating with samples, aiming to control for MS variability (MS QC). Finally, the 25 DDA pools were consolidated in a single aliquot and injected 12 times at the end of the batch (replicate QC) [21].

### 2.7. MS Data Processing

MaxQuant software (v. 2.2.0.0) was used to search spectra in .raw files from DDA data against the human proteome (Uniprot, 20594 entries downloaded in February 2023). The protein detection and quantification criteria included at least two peptides (one being a unique peptide) from specific trypsin hydrolysis (up to two missed cleavages were allowed). Methionine oxidation and cysteine carbamidomethylation were selected as variable and fixed modifications, respectively. MaxQuant .msms file was converted to a .dlib file using EncyclopeDIA (v.1.12.31), employed as the library for DIA. For data processing, DIA .raw files were converted to .mzML using MSConvert (v.3.0.22297) and used within EncyclopeDIA software (v.1.12.31) to quantify proteins [20]. Software configurations included the selection of 4 to 6 transitions per peptide, mass tolerance set at a resolution of 30,000, and Percolator v.3-01. Protein intensities were TIC normalized. The output of protein intensities was used for quantification, with a specific protein being considered for quantification if coefficients of variation (CVs) lower than 25% were obtained for this protein in at least 2 out of 3 QCs (Digestion, MS or replicate QCs), after outlier removal using Grubbs’ test. Protein intensities equal to zero were considered missing values. Keratins were considered contaminants from sample preparation and were also excluded from analysis.

### 2.8. Statistical Analysis

The Shapiro–Wilk test was used to assess the normality of quantitative variables. Non-parametric data were represented by median and interquartile ranges (25% and 75%), and comparison between the two groups was performed using the Mann–Whitney test or the Kruskal–Wallis test (when comparing more than two samples), with the Bonferroni post-test. A significance level of *p* < 0.05 was considered. IBM^®^ SPSS Statistics (version 29.0), GraphPad Prism (version 5.04) for Windows, and Microsoft^®^ Excel for Mac (version 16.77.1) software were used for data tabulation and analysis. Heat maps were constructed by using MetaboAnalyst 6.0.

## 3. Results

Age, body mass index (BMI), and clinical characteristics of groups are presented in Table 1. Women with BC were older than CTs, as well as the percentage of women in the post-menopausal state. Body mass index (BMI) did not significantly differ between the two groups. Luminal A (LA) and LB types constituted the majority of tumors (68%), with most cases classified as stages I and II; 76%). HER2 and TNBC constituted 17% and 14% of the BC study population. Further stratification based on the molecular types of BC revealed similarities across types LA, LB, HER2, and TNBC regarding age distribution, BMI, and menopausal status (Table 2).

Plasma levels of TC, TG, HDLc, LDLc, nonHDLc, and the ratios of TC/apoB and TG/HDLc were similar between CT and BC groups (Table 3) as well as among the molecular types of the disease (Table 4).

The proteins associated with HDL were analyzed through DIA proteomics, employing nano-scale LC/MS/MS. Tables with protein intensities and information regarding samples’ information have been included in the MassIVE repository. Metadata files including the list of all identified proteins, peptides with retention times, and quantitative information are depicted in Appendix A.

Eighty-eight proteins were associated with HDL particles in both the CT and BC groups. Among these, 26 proteins were quantifiable (Table 5), with 6 showing lower levels in the BC group compared to the CT group (PON1, apoA-4, TTR, CLEC3B, IGHM, and SAA2) (Figure 1A–F), and only 1 exhibiting higher levels in BC compared to the CT group (CNDP1) (Figure 1G).

A heat map analysis of proteins present in HDL of CT and BC cases is depicted in Figure 2. The ROC analysis evidenced the discriminative capacity for CNPD1 (AUC = 0.5890; *p* = 0.0176), IGHM (AUC = 0.5846; *p* = 0.0241), and SAA2 (AUC = 0.5920; *p* = 0.0146)].

According to the molecular type of BC, 14 proteins were differentially abundant. From these, 4 were reduced (apoA-1; apoC-2; apoA-2 and apoA-4) (Figure 3A–D), while 10 were higher in TNBC compared to LA, LB, and HER-2 (C3; CFB; IGLC2/3; GC; PLG; serpin 3; IGHC1; C9; LRG 1, and C4B) (Figure 4A–J).

A heat map analysis of proteins expressed in the HDL of BC cases is depicted in Figure 5, and Table 6 shows the discriminative capacity of the four under-expressed and ten overexpressed proteins in HDL between TNBC cases and LA, LB, or HER2 types according to ROC analysis.

The analysis between the clinical stages of the disease revealed a decrease in the protein CLEC3B (Figure 6A) and an increase in five proteins (A2M; HPR; LCAT; SAA1; and IGHM) in the HDL of women in stages III + IV grouped compared to those in stages I + II (Figure 6B–F). Nonetheless, by ROC analysis, only A2M (AUC = 0.6214; *p* = 0.0204) and HPR (AUC = 0.6230; *p* = 0.0187) displayed a good discriminatory capacity between stages of the disease. The heat map depicted in Figure 7 shows differentially abundant proteins in the stages of BC.

## 4. Discussion

Traditional risk factors for BC have been extensively studied, and identifying new contributors to the disease could aid in its prevention and clinical management. In this study, a distinct profile of HDL proteomics was observed between newly diagnosed BC cases and matched controls, especially in TNBC compared to other molecular types of BC. Furthermore, the levels of proteins in HDL varied across the clinical stages of the disease.

Quantification of HDL proteins is mainly obtained by DDA proteomics. However, this untargeted method is stochastic and may suffer from technical variabilities. Targeted strategies enable the precise quantification of low-abundance proteins with exceptional accuracy, sensitivity, and reproducibility. Targeted proteomics, represented by selected reaction monitoring (SRM) is the gold standard for MS quantification. Another targeted quantification strategy, called parallel reaction monitoring (PRM) showed comparable analytical performance to that obtained by SRM to quantify HDL proteome [17]. Data-independent acquisition (DIA), used in the present study, is a relatively recent quantification strategy in MS-based proteomics, that combines the benefits of both untargeted and targeted approaches. This method enables samples to be acquired once and analyzed multiple times in silico, as all fragment ions are recorded and assessed. Importantly, it has been demonstrated that both DIA and PRM are equally effective in differentiating HDL subclass proteomes, with comparable analytical performances and a high level of agreement [18]. Moreover, the variability of DIA for HDL proteomics was determined to be low, making this an excellent quantitative method [20].

According to their main function, antioxidant proteins (PON-1, CNPD1, and apo A4), and transport-related proteins (IGHM and TTR) were decreased; and the inflammatory protein, SAA2, and the tissue remodeling protein, CLEC3B, were increased in HDL of BC compared to that of CTs. Interestingly, CNPD1, SAA2, and IGHM had a better discriminatory power between CT and BC cases.

Orkuturlar et al. (2018) evaluated 40 subjects with BC and 33 women as controls, demonstrating that serum concentrations of PON1 and arylesterase (ARE) were significantly lower in those subjects who required neoadjuvant chemotherapy [22]. On the other hand, Arenas et al. (2017) found an increase in PON1 activity in 200 women with BC after radiotherapy. This increase in PON1 may have clinical implications and suggests an improvement in the overall clinical condition [23]. The enzyme carnosine dipeptidase 1 (CNDP1), also known as beta-Ala-His-dipeptidase, is encoded by the *CNDP1* gene in humans. It has beneficial effects on various chronic diseases due to its anti-inflammatory, antioxidant, and antiglycation properties, although human studies need better validation [24]. ApoA-4 exerts antiatherogenic actions, complemented by anti-inflammatory [25] and antioxidant properties [26,27]. ApoA-4 plasma levels were higher in *BRCA* mutation carriers without BC in contrast to those with BC [28]. Transthyretin (TTR) is reduced in BC serum regardless of chemotherapy [29] and is associated with tamoxifen resistance [30].

Immunoglobulin M (IGHM) complexes play an important role in the immune system, although its implication in BC is not clear. In lung adenocarcinoma, IGHM was in the top five marker genes of B cells in single-cell sequencing data [31].

Although, in the present investigation, an increase in CLEC3B or soluble tetranectin in the HDL of BC was observed, it is reported a reduction in serum concentrations of CLEC3B in patients with BC [32,33,34,35,36] and elevated serum concentrations indicated a more favorable prognosis. Additionally, tetranectin concentrations in plasma were considered a significant predictor of the response to chemotherapy in patients with metastatic BC [37].

SAA2, which was higher in the HDL of BC, is described as elevated in TNBC compared to other molecular types and is higher in subjects with ER-negative breast tumors compared to those with ER-positive tumors. High concentrations of SAA2 are associated with a survival time of less than one year in women with invasive ductal carcinoma, making it a useful marker for BC recurrence [38]. Moreover, BC in stages II, III, and IV had higher concentrations of SAA2 compared to the control group and stage I. Additionally, patients with BC who have lymph node involvement or distant metastases show elevated concentrations of SAA2 [39].

A higher number of differently abundant proteins in HDL were found when comparing TNBC with other molecular types of BC. Decreased levels of apoA-1, apoA-2, apoC-2, and apoC-4 were observed in TNBC, all proteins related to HDL metabolism and function. ApoA-1 and apoA-2 are the primary structural apolipoproteins of HDL involved in reverse cholesterol transport. ApoA-1 plays a crucial role in HDL biogenesis by promoting the uptake of excess cell cholesterol and phospholipids, via ATP binding cassette transporter A-1 (ABCA-1) to form pre-beta HDL, activating lecithin cholesterol acyltransferase (LCAT), and inhibiting inflammation and immune response [40]. Reductions in apoA-1 plasma levels are related to the onset and progression of cancer [41,42] and BC [43]. Liu et al. (2019) showed that apoA-1 is a reliable marker for distinguishing BC subjects with intraocular metastasis from those without this condition [44].

ApoA-2 constitutes approximately 20% of the protein composition of HDL, making it the second most abundant protein in this lipoprotein. It plays an important role in regulating the activity of key components such as LCAT, cholesterol ester transfer protein (CETP), and lipoprotein lipase [45]. ApoC-2 and apoC-4, which were also reduced in BC cases, regulate HDL generation according to the modulation of triglyceridemia. The concentration of apoC-2 was lower in hormone receptor-negative BC tumors with a high proliferative index [46]. On the other hand, the increase in complement system proteins (C3, C9, C4B, and CFB), serine protease inhibitors (serpin 3, LRG1, and PLG), and transport proteins (IGLC2/3, IGHG1, and GC) may interfere with reverse cholesterol transport, and increase inflammatory, oxidative, and immune responses. In BC, IGHG1 level is increased, triggering the activation of protein kinase B (AKT) and vascular endothelial growth factor (VEGF) signaling, which promotes increased cell proliferation, invasion, and angiogenesis. Zhang et al. (2023) demonstrated that silencing IGHG1 in C57BL/6j mice suppressed the neoplastic characteristics of cancer cells in vitro and reduced tumor growth [47].

TNBC is recognized as a more aggressive type of BC characterized by the absence of specific and effective therapies and a low survival rate. Particularly, in TNBC, higher plasma cholesterol and TG levels are reported and may help drive lipids for tumor replication and metastasis [2]. HDL can alter the tumor microenvironment by modulating lipid content, especially by mediating excess cholesterol removal by ABCA-1, ATP-binding cassette transporter G-1 (ABCG-1), and scavenger receptor class B type 1 (SR-B1) through reverse cholesterol transport. This seems crucial to limit the amount of cholesterol and oxysterols necessary for cell replication, epithelial–mesenchymal transition, and metastasis [48]. 27- and 25-hydroxycholesterol are selective estrogen receptor modulators involved in an LXR-mediated metastasis potential in BC [49]. Notably, the ability of HDL to remove cell cholesterol is impaired in more advanced stages (III and IV) of BC (mainly represented by TNBC cases), which correlates with a reduced content of cholesterol and phospholipids in HDL particles in BC subjects. Additionally, the levels of 25- and 27-hydroxycholesterol, which contribute to an alternative route of reverse cholesterol transport, are lower in the HDL of women with BC in comparison to controls [3].

Moreover, HDL can mitigate oxidative stress by carrying PON1, other antioxidant enzymes, and apoA-1. In this sense, it was recently demonstrated that, different from expected, HDL from TNBC cases have a greater ability to inhibit LDL oxidation by enhancing the lag time for copper sulfate-mediated LDL oxidation in vitro [50]. In addition, HDL from stages III and IV has a greater ability to inhibit inflammation in macrophages, which was related to the enhanced expression of miR223 and miR375 in HDL [51]. These observations draw attention to the effect of HDL nourishing TNBC development and aggressiveness.

Interestingly, HDL from stages III and IV grouped BC differed in its proteomics in comparison to stages I and II: alpha2-macroglobulin (A2M) and soluble tetranectin (CLEC3B) were reduced, and HPR, LCAT, SAA1, and IGHM were increased. In accordance with our observation, Liang et al. (2006) found a significant decrease in alpha-2-macroglobulin (A2M) in BC [52]. On the other hand, this protein was shown to be increased in the plasma and tissue of TNBC [53]. Also, a higher concentration of serum amyloid A1 (SAA1) was observed in BC and linked to a worse prognosis of the disease [54] by mediating cell proliferation, inflammation, angiogenesis, and metastasis [55,56]. In opposition to our findings, the haptoglobin-related protein (HPR) was also found to be elevated in the early stages of BC, being considered an independent prognostic factor for disease recurrence [57].

The exact interplay between HDL and tumors is difficult to assess, and HDL proteomics and composition accessed from plasma may be a consequence rather than a cause of tumor development. It is important to consider that HDL can be modulated by tumor cells and immune cells, such as macrophages, which infiltrate the tumor microenvironment. These cells can uptake modified low-density lipoproteins, leading to disturbances in intracellular lipid content, which can elicit oxidative and inflammatory stress. This stress can modulate neighboring cells through paracrine signaling. Consequently, the HDL functionality which is crucial in limiting excess sterols and balancing oxidation and inflammation can be damaged by the tumor. These interactions among different cell types in the tumor and HDL may explain the controversial results regarding HDLc levels and BC development [4,5,7,8,9].

Proteomic analyses have been conducted on various samples, including tissues, whole plasma, and urine, to better understand the pathophysiology of BC. The analysis of formalin-fixed paraffin-embedded tissue specimens obtained from BC surgery revealed a distinct proteome with a higher expression of immune response proteins related to better clinical outcomes. In a subgroup of TNBC cases, four proteome clusters were found to be related to clinical outcomes. Particularly, in cluster 1, a higher expression of immune response proteins, antigen processing, and presentation, and type I and II interferon signaling processes were associated with better recurrence-free survival and overall survival [58]. A different profile of proteins expressed in urinary vesicles was recently described in a small cohort of BC cases compared to controls. Proteins related to cell proliferation, migration, and survival were identified as potential biomarkers for the screening and diagnosis of BC [59]. In another study, the examination of the proteomic profile of serum extracellular vesicles from subjects with BC showed that TALDO1 a rate-limiting enzyme of the pentose phosphate pathway was found to be a biomarker of tumor metastasis [60]. Given the unique aspects of TNBC resistance to standard therapies and its associated poor prognosis, there is a critical need for the thorough validation of biomarkers and underlying mechanisms specific to this BC type.

It is important to note that HDL shares proteins with other lipoproteins in plasma, but certain proteins function differently depending on the specific lipoprotein they are associated with [61]. Moreover, some proteins exist in plasma both as an HDL-bound form and in a lipid-free fraction, and they also display different properties [62,63]. Thus, the determination of the HDL proteome adds a new piece of information to BC biology.

## 5. Conclusions

Findings from the present investigation dealing with a large cohort reveal a distinct profile of proteins present in HDL particles among TNBC cases, demonstrating significant discriminatory power compared to other molecular types of BC. Moreover, the HDL proteome showed discriminatory abilities across different clinical stages of the disease. A prospective follow-up study will be essential to validate the prognostic value of HDL functionality and proteomics in predicting outcomes in BC and optimizing emerging chemo- and immunotherapies.

## Figures and Tables

**Figure 1 cells-13-01327-f001:**
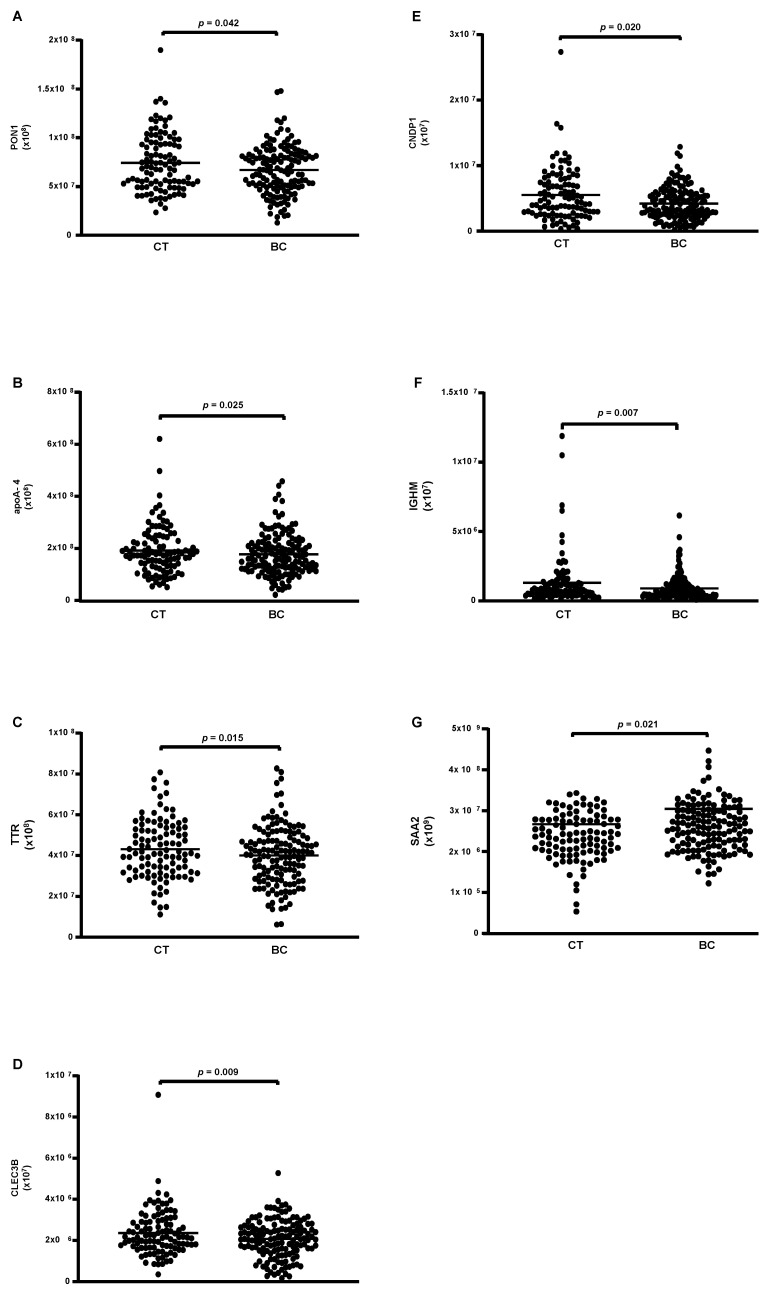
Proteins are differentially abundant in HDL of CT and BC cases. HDL was isolated through ultracentrifugation on a discontinuous density gradient (D = 1.063–1.21 g/mL). Five micrograms of HDL protein from individuals with BC (n = 141) and CT (n = 143) were trypsin-digested. Samples were desalted, and after MS, the identified proteins were quantified by DIA proteomics. Comparisons were made by the Mann–Whitney test (median and interquartile range 25–75%). CT = control; BC = breast cancer. (**A**) paraoxonase 1 (PON1); (**B**) apolipoprotein A-4; (**C**) transthyretin (TTR); (**D**) tetranectin (CLEC3B); (**E**) carnosine dipeptidase 1 (CNDP1); (**F**) immunoglobulin Mu heavy chain constant region (IGHM); (**G**) serum amyloid A type 2 (SAA2).

**Figure 2 cells-13-01327-f002:**
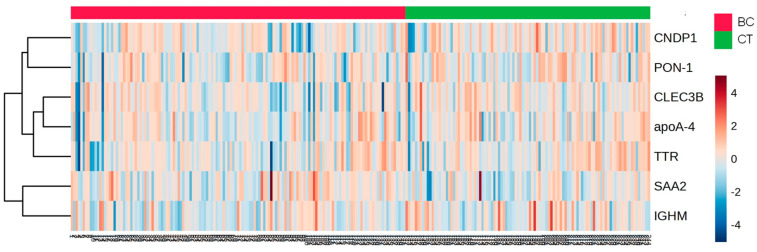
Heat map analysis of HDL proteome in CT and BC cases. HDL was isolated through ultracentrifugation on a discontinuous density gradient (D = 1.063–1.21 g/mL). Five micrograms of HDL protein from individuals with BC (n = 141) and CT (n = 143) were trypsin-digested. Samples were desalted, and after MS, the identified proteins were quantified by DIA proteomics. The heat map was constructed by using MetaboAnalyst 6.0. CT = control; BC = breast cancer.

**Figure 3 cells-13-01327-f003:**
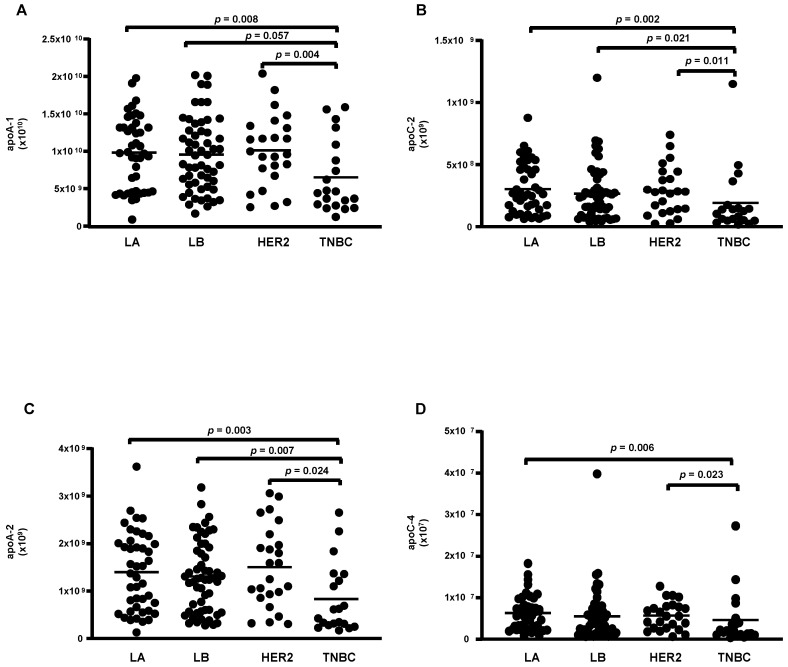
Proteins downregulated in HDL proteomes of TNBC cases. HDL was isolated through ultracentrifugation on a discontinuous density gradient (D = 1.063–1.21 g/mL). Five micrograms of HDL protein from individuals with BC (n = 141) and CT (n = 143) were trypsin-digested. Samples were desalted, and after MS, the identified proteins were quantified by DIA proteomics. Comparisons among the molecular types of BC were made by one-way ANOVA (median and interquartile range 25–75%). LA = luminal A; LB = Luminal B; TNBC = triple-negative breast cancer. (**A**) apolipoprotein A-1 (apoA-1); (**B**) apolipoprotein C-2 (apoC-2); (**C**) apolipoprotein A-2 (apoA-2); (**D**) apolipoprotein C-4 (apoC-4).

**Figure 4 cells-13-01327-f004:**
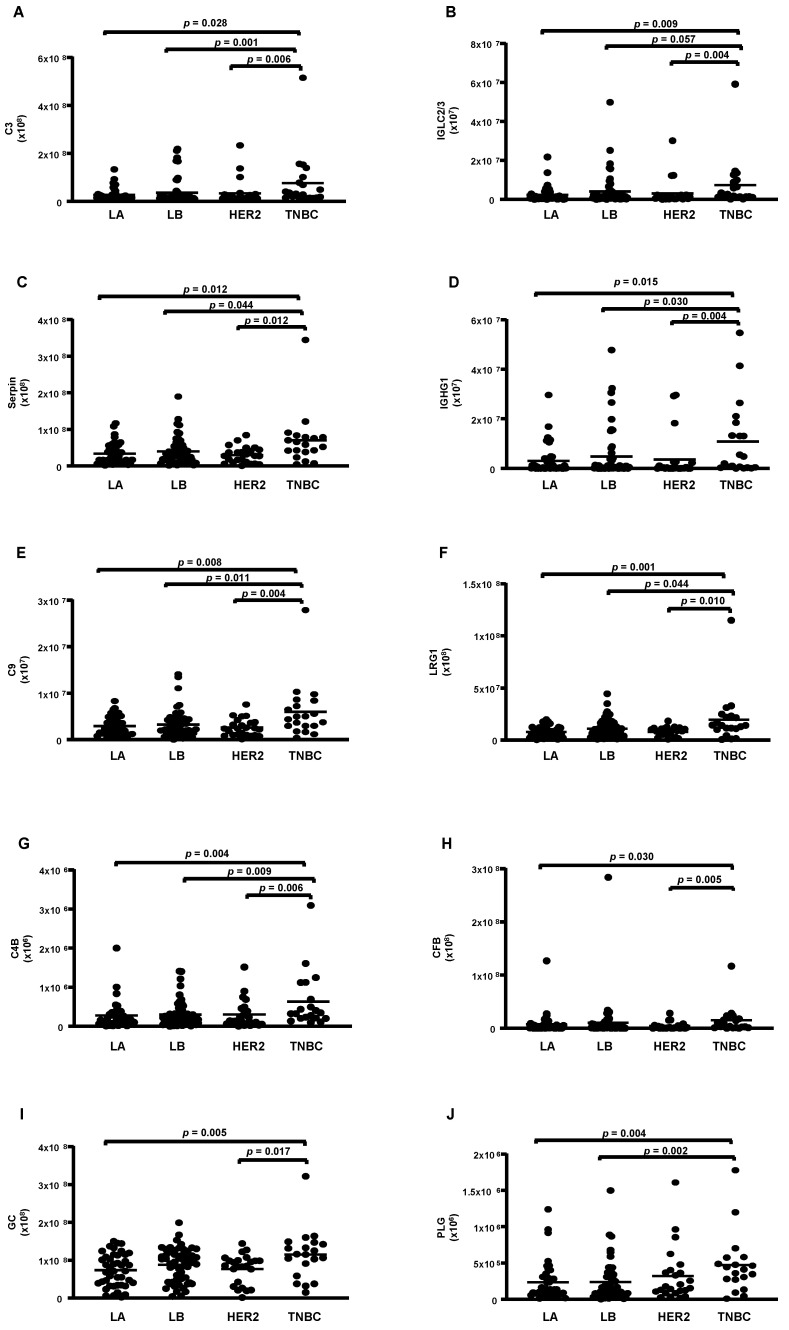
Proteins upregulated in HDL proteomes of TNBC cases. HDL was isolated through ultracentrifugation on a discontinuous density gradient (D = 1.063–1.21 g/mL). Five micrograms of HDL protein from individuals with BC (n = 141) and CT (n = 143) were trypsin-digested. Samples were desalted, and after MS, the identified proteins were quantified by DIA proteomics. Comparisons among the molecular types of BC were made by one-way ANOVA (median and interquartile range 25–75%). LA = luminal A; LB = Luminal B; TNBC = triple-negative breast cancer. (**A**) complement C3 (C3); (**B**) immunoglobulin lambda constant 2/3 (IGLC2/3); (**C**) serpin; (**D**) immunoglobulin heavy constant gamma 1 (IGHG1); (**E**) complement C9 (C9); (**F**) leucin-rich alpha2-glycoprotein 1 (LRG1); (**G**) complement 4B (C4B); (**H**) complement factor B (CFB); (**I**) vitamin D-binding protein or group specific component (CFB); (**J**) plasminogen (PLG).

**Figure 5 cells-13-01327-f005:**
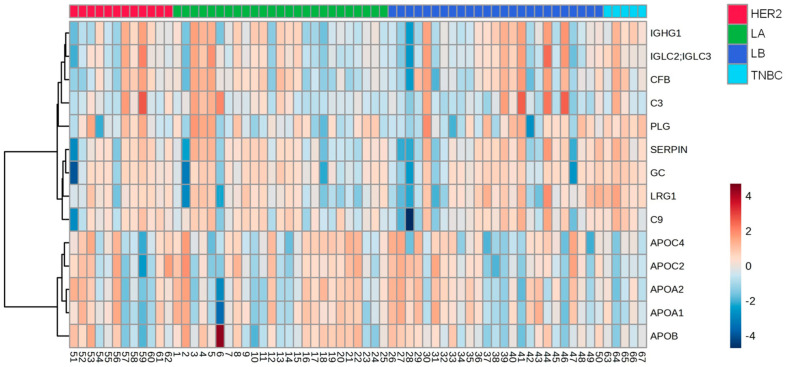
Heat map analysis of HDL proteomes in BC cases according to the molecular type of the disease. HDL was isolated through ultracentrifugation on a discontinuous density gradient (D = 1.063–1.21 g/mL). Five micrograms of HDL protein from individuals with BC (n = 141) and CTs (n = 143) were trypsin-digested. Samples were desalted, and after MS, the identified proteins were quantified by DIA proteomics. The heat map was constructed by using MetaboAnalyst 6.0. LA = luminal A; LB = Luminal B; TNBC = triple-negative breast cancer.

**Figure 6 cells-13-01327-f006:**
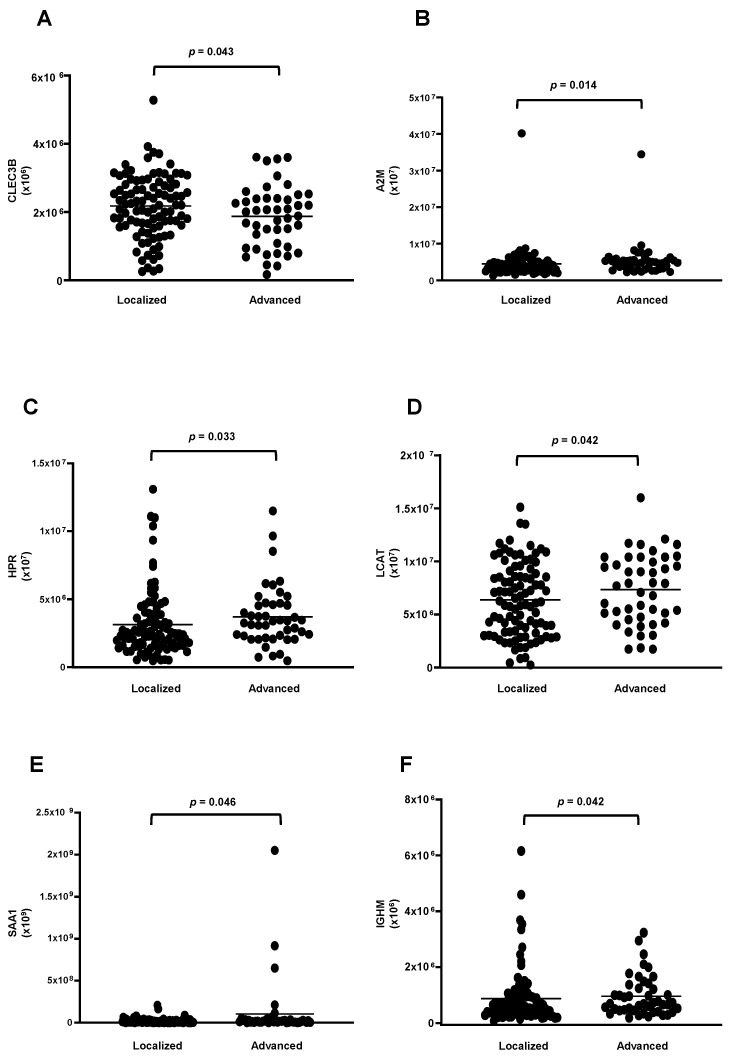
Proteins downregulated and upregulated in HDL proteomes of stages III and IV of BC. HDL was isolated through ultracentrifugation on a discontinuous density gradient (D = 1.063–1.21 g/mL). Five micrograms of HDL protein from individuals with BC (n = 141) and CT (n = 143) were trypsin-digested. Samples were desalted, and after MS, the identified proteins were quantified by DIA proteomics. Comparisons between stages I + II and III + IV were made by the Mann–Whitney test (median and interquartile range of 25–75%). (**A**) tetranectin (CLEC3B); (**B**) alpha2-macroglobulin (A2M); (**C**) haptoglobin -related protein (HPR); (**D**) lecithin cholesterol acyltransferase (LCAT); (**E**) serum amyloid A type 1 (SAA1); (**F**) immunoglobulin Mu heavy chain constant region (IGHM).

**Figure 7 cells-13-01327-f007:**
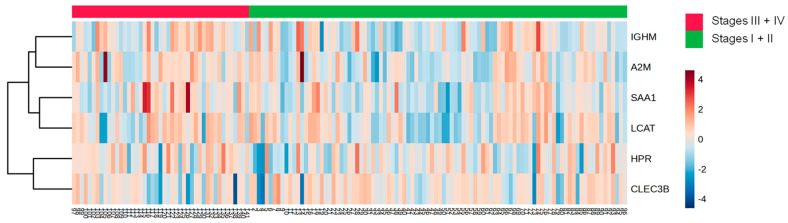
Heat map analysis of HDL proteomes according to clinical stages of BC. HDL was isolated through ultracentrifugation on a discontinuous density gradient (D = 1.063–1.21 g/mL). Five micrograms of HDL protein from individuals with BC (n = 141) and CT (n = 143) were trypsin-digested. Samples were desalted, and after MS, the identified proteins were quantified by DIA proteomics. The heat map was constructed by using MetaboAnalyst 6.0.

**Table 1 cells-13-01327-t001:** Age, BMI, and menopausal status in CT, BC groups, and molecular types of BC.

	CT		BC		*p*
n	143		141		
Age (year)	50 (38–59)		56 (49–63)		<0.0001
BMI (kg/m^2^)	27 (24–31)		27 (24–31)		0.6391
	n	%	n	%	
Pre-menopause	72	50	49	34	
Post-menopause	69	48	92	65	

CT = control; BC = breast cancer; BMI = body mass index.

**Table 2 cells-13-01327-t002:** Age, BMI, and menopausal status in molecular types of BC.

	LA		LB		HER2		TNBC		*p*
n	43		54		24		20		ns
Age (year)	61 (52–65)		53 (46–61)		57 (52–61)		56 (41–61)		ns
BMI (kg/m^2^)	27 (24–30)		27 (24–30)		27 (24–31)		28 (23–32)		ns
	n	%	n	%	n	%	n	%	
Pre-menopause	10	23	26	48	5	20	8	40	
Post-menopause	33	46	28	51	19	79	12	60	

BMI = body mass index; LA = luminal A; LB = luminal B; TNBC = triple-negative breast cancer; ns = not significant.

**Table 3 cells-13-01327-t003:** Plasma lipid profile and lipid ratios of CT and BC groups.

	CT	BC	*p*
n	143	141	
TC (mg/dL)	173 (153–202)	186 (158–212)	0.4272
TG (mg/dL)	84 (59–119)	92 (68–115)	0.5551
apoB (mg/dL)	110 (89–137)	110 (81–141)	0.5615
HDLc (mg/dL)	42 (35–51)	41 (34–49)	0.6909
LDLc (mg/dL)	112 (94–134)	117 (98–144)	0.6348
Non-HDLc (mg/dL)	129 (129–158)	135 (116–165)	0.5720
TC/apoB	1.5 (1.3–1.9)	1.7 (1.3–2.0)	0.1595
TG/HDLc	1.8 (1.3–3.0)	2.2 (1.4–3.2)	0.5864

CT = control; BC = breast cancer; TC = total cholesterol; TG = triglycerides; apoB = apolipoprotein B; HDLc = HDL cholesterol; LDLc = LDL cholesterol; non-HDLc = non-HDL cholesterol.

**Table 4 cells-13-01327-t004:** Plasma lipid profile and lipid ratios in molecular types of BC.

	LA	LB	HER2	TNBC	*p*
n	43	54	24	20	
TC (mg/dL)	195 (162–232)	175 (156–209)	173 (153–197)	191 (164–210)	ns
TG (mg/dL)	104 (66–139)	84 (68–112)	85 (67–100)	99 (82–115)	ns
apoB (mg/dL)	129 (91–148)	98 (79–126)	103 (80–137)	134 (95–173)	ns
HDLc (mg/dL)	41 (34–48)	42 (32–53)	45 (35–51)	40 (34–44)	ns
LDLc (mg/dL)	122 (103–159)	111 (96–140)	116 (84–137)	128 (97–149)	ns
Non-HDLc (mg/dL)	150 (122–190)	130 (108–162)	133 (103–153)	148 (129–175)	ns
TC/apoB	1.7 (1.3–2.0)	1.8 (1.5–2.4)	1.8 (1.3–1.9)	1.4 (1.1–1.9)	ns
TG/HDLc	2.2 (1.5–3.1)	2.0 (1.3–3.2)	1.6 (1.2–3.1)	2.4 (1.8–3.3)	ns

LA = luminal A; LB = luminal B; TNBC = triple-negative breast cancer; TC = total cholesterol; TG = triglycerides; apoB = apolipoprotein B; HDLc = HDL cholesterol; LDLc = LDL cholesterol; non-HDLc = non-HDL cholesterol; ns = not significant.

**Table 5 cells-13-01327-t005:** Selected and quantified proteins in HDL proteomics, divided by their main function.

Abbreviations	Proteins
Lipid metabolism	
ApoA-4	Apolipoproteins A-4
ApoA-1	Apolipoprotein A-1
Apo-C-2	Apolipoprotein C-2
ApoC-4	Apolipoprotein C-4
LPA	Apolipoprotein (a)
LCAT	Lecithin cholesterol acyltransferase
Inflammatory acute phase protein	
SAA1	Serum Amyloid A type 1
SAA2	Serum Amyloid A type 2
HPR	Haptoglobin-related protein
Complement system	
C3	Complement C3
C9	Complement C9
CFB	Complement Factor B
C4B	Complement 4b
Antioxidants	
PON1	Paraoxonase-1
CNDP1	Carnosine dipeptidase 1
Transport proteins	
GC	Vitamin D-binding protein or group-specific component
TTR	Transthyretin
IGHM	Immunoglobulin Mu heavy chain constant region
IGLC2/3	Immunoglobulin lambda constant 2/3
IGHG1	Immunoglobulin heavy constant gamma 1
Tissue remodeling	
CLEC3B	Tetranectin
Serine protease inhibitors	
SERPIN 3	Serpin 3
PLG	Plasminogen
LRG1	Leucine-rich alpha-2 glycoprotein 1
A2M	Alpha2-macroglobulin

**Table 6 cells-13-01327-t006:** Discriminative capacity of proteins with lower or higher abundance in TNBC HDL compared to other molecular types of BC.

Lower Expression in TNBC				
Protein		vs. LA	vs. LB	vs. HER2
ApoA-1	AUC *p*	0.7017 0.0104	0.6936 0.0107	0.7104 0.0173
ApoC-2	AUC *p*	0.7366 0.0027	0.6891 0.0127	ns
ApoA-2	AUC *p*	0.7279 0.0038	0.7091 0.0059	0.7438 0.0058
ApoC-4	AUC *p*	0.7134 0.0067	ns	0.6958 0.0267
**Higher expression in TNBC**				
Protein		vs. LA	vs. LB	vs. HER2
C3	AUC *p*	0.6965 0.0126	0.7259 0.0029	0.7365 0.0075
IGLC2/3	AUC *p*	0.7140 0.0066	0.6450 0.0560	0.7375 0.0072
Serpin	AUC *p*	0.7240 0.0052	0.6885 0.0148	0.7588 0.0039
IGHG1	AUC *p*	0.6988 0.0116	0.6676 0.0276	0.7333 0.0083
C9	AUC *p*	0.7058 0.0090	0.6973 0.0093	0.7458 0.0054
LRG1	AUC *p*	0.7360 0.0027	0.6505 0.0474	0.7573 0.0036
C4B	AUC *p*	0.7302 0.0035	0.7091 0.0059	0.7042 0.0209
CFB	AUC *p*	0.6779 0.0239	ns	0.7313 0.0089
GC	AUC *p*	0.7110 0.0074	ns	0.7281 0.0099
PLG	AUC *p*	0.7295 0.0042	ns	ns

ns = not significant.

## Data Availability

All data reported are included in the manuscript and raw data can be kindly shared upon personal request to the corresponding author, M.P. (m.passarelli@fm.usp.br).

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
