# Peer review of "Proteomic Profiling of HDL in Newly Diagnosed Breast Cancer Based on Tumor Molecular Classification and Clinical Stage of Disease"

_cells, 2024, doi:10.3390/cells13161327_

Round 1

Reviewer 1 Report

Comments and Suggestions for Authors

In this study, authors reported differentially expressed proteins isolated from HDL samples collected from the plasma of BC patients using a shotgun proteomic approach.  Although the paper is interesting and can serve as a proof of concept for the use of advanced mass spectrometry-based methodology as a tool to identify specific biomarkers in breast cancer, I have some suggestions and corrections, and the paper would benefit from additional work before publication.

1.     Introduction: The authors had a detailed explanation of different classes of breast cancer types occurring in patients. However, it would be beneficial to include some more introduction on the shotgun proteomic approach, especially focussing on HDL proteomics to guide the reader through why this is important in this field.

2.     Line 126: HDL isolation – Please be more specific on the conditions of the ultracentrifugation including the make, model, speed etc. Can you also clarify, after ultracentrifugation – do HDL proteins are isolated as a pellet? Do you wash it with any buffer? Or just remove the supernatant and store it at -80 °C?

3.     Line 131: HDL proteomics – Please include the basic methods used for proteomic digestion – is it done by FASP or in-solution digestion? Please include relevant citations for these methods. Reference [17] corresponds to Santana, Monique de Fatima Mello et al. The paper didn’t have any proteomics included.

4.     Authors mentioned that Angiotensin peptide is used as the global internal standard in this study. Can you please clarify if this standard was also used for checking the instrument stability, and also used for establishing the lower limit of quantitation for each protein analyzed using PRM? 

5.     Line 146 – Authors deposited all raw mass spec files on MASSIVE. Most of the proteomics papers include other metadata files including the list of all identified proteins, peptides with retention times, and quantitative information in the form of Excel or as supplementary tables. Please include this information in the manuscript.  

6.     Line 154 – Statistical analysis – The authors mentioned that PRM data was analyzed using Skyline software. Nothing has been included in the data analysis section – on how the integration of the peptides was done and how was the data normalized etc? This is an important part of proteomic analysis. Including all details improves the paper.

7.     The authors haven’t included the UniProt FASTA file details used for the analysis of the PRM data. Please include the date of accession of these details. Can you please include the details on how the library of specific proteins has been constructed for the analysis? How many peptides were considered, and how the selection was made, and what possible cut-offs were used to generate the library.

8.     Results: The figures shown in the paper need to include more information. It wasn’t clear what parameters were considered for the statistical differences. – are those peak areas, or fold change relative abundance or absolute quantities of the proteins? Including data analysis details as requested above would be clearer on what is considered for the statistical significance.

9.     Please format citations in the discussion and keep them consistent. For example, in line no-290 – Arenas           M et al – remove M. In Line 336 – the year was missing. So, either remove the year or keep them.

10.  Authors had discussed the results of dysregulated protein expressions in urinary and serum vesicles in a paragraph (line – 384 to 388). Though this is relevant to BC, this is not adding up to the aim and hypothesis of the current paper. Please discuss this further and try to connect with the paper, or is this suggesting any future studies? Not clear.

11.  It would be good if authors could discuss the advantage of PRM over DIA proteomics, and how this significantly improves the understanding of HDL. The proteins discussed in this study could also be identified from undepleted plasma samples – so, can you elaborate on why it is important to do HDL proteomics?

12.  Conclusion section is missing.

13. Please check for minor editing errors, in the document. 

Author Response

In this study, authors reported differentially expressed proteins isolated from HDL samples collected from the plasma of BC patients using a shotgun proteomic approach.  Although the paper is interesting and can serve as a proof of concept for the use of advanced mass spectrometry-based methodology as a tool to identify specific biomarkers in breast cancer, I have some suggestions and corrections, and the paper would benefit from additional work before publication.

The authors express their gratitude to the reviewer for their insightful comments and valuable suggestions, which have greatly contributed to enhancing the quality of the manuscript. Please, find below a detailed response addressing each comment. The revisions have been highlighted in red in the latest version of the manuscript.

  1. Introduction: The authors had a detailed explanation of different classes of breast cancer types occurring in patients. However, it would be beneficial to include some more introduction on the shotgun proteomic approach, especially focussing on HDL proteomics to guide the reader through why this is important in this field.

The authors have added a sentence to the introduction discussing the potential of using HDL proteomics to better understand BC pathophysiology. (line 83) “The analysis of the HDL proteome enabled the identification of hundreds of proteins associated with various HDL subfractions, modulating different functional properties of these lipoproteins. Additionally, HDL proteomics can support the identification of biomarkers related to disease and therapeutic responses”.

In the discussion, another sentence has been added that addresses different approaches to measuring HDL proteomics and provides a brief comparison among them. (line 339) “Quantification of HDL proteins is mainly obtained by data-dependent acquisition (DDA) proteomics. However, this untargeted method is stochastic and may suffer from technical variabilities. Targeted strategies enable precise quantification of low-abundance proteins with exceptional accuracy, sensitivity, and reproducibility. Targeted proteomics, represented by selected reaction monitoring (SRM) is the gold standard for MS quantification. Another targeted quantification strategy, called parallel reaction monitoring (PRM) showed comparable analytical performance to that obtained by SRM to quantify HDL proteome [17]. Data-independent acquisition (DIA), used in the present study, is a relatively recent quantification strategy in MS-based proteomics, that combines the benefits of both untargeted and targeted approaches. This method enables samples to be acquired once and analyzed multiple times in silico, as all fragment ions are recorded and assessed. Importantly, it has been demonstrated that both DIA and PRM are equally effective in differentiating HDL subclass proteomes, with comparable analytical performances and a high level of agreement [18]. Moreover, the variability of DIA for HDL proteomics was determined to be low, making this an excellent quantitative method [20].  

  1. Line 126: HDL isolation – Please be more specific on the conditions of the ultracentrifugation including the make, model, speed etc. Can you also clarify, after ultracentrifugation – do HDL proteins are isolated as a pellet? Do you wash it with any buffer? Or just remove the supernatant and store it at -80 °C?

A more detailed explanation of ultracentrifugation and HDL processing has been added (line 132), as follows: “High-density lipoprotein (HDL; D = 1.063–1.21 g/mL) was isolated from plasma by discontinuous density ultracentrifugation for 24 h, 4 °C, 100,000 g, (SW40 rotor; L80-Beckman ultracentrifuge). Lipoproteins were carefully removed by vacuum and HDL fraction was stored at − 80 °C in a 5% saccharose solution. HDL composition in apoA-1 was determined by immunoturbidimetry (Randox Lab. Ltd. Crumlin, UK) and in lipids (TC, TG, and PL) by enzymatic techniques (Roche Diagnostics, SP, Brazil)”.

  1. Line 131: HDL proteomics – Please include the basic methods used for proteomic digestion – is it done by FASP or in-solution digestion? Please include relevant citations for these methods.

The authors apologize for the error, as the methodology described was based on our previous publication on HDL proteomic analysis by PRM. A solution digestion was utilized. In the new version of the manuscript, the method used in the present investigation for HDL proteomic assessment by DIA is described in detail (lines 138 to 201).

“HDL digestion for proteomics

The Bradford assay (Bio-Rad, Hercules, USA) determined total protein concentration in isolated HDL. Five micrograms of HDL-protein were diluted in 50 mM ammonium bicarbonate buffer (Sigma-Aldrich, St. Luis, USA) containing 0.01% ProteaseMAX MS-compatible surfactant (Promega). Proteins were reduced with 5 mM dithiothreitol (Bio-Rad), for 1h at 37 °C, alkylated with 15 mM iodoacetamide (Bio-Rad) for 30 min and excess iodoacetamide was quenched using 2.5 mM dithiothreitol for 15 min at room temperature RT. Proteins were then digested with trypsin [1:40, w:w, trypsin (Promega, Madison, USA): HDL proteins] for 4h at 37 °C. A second aliquot of trypsin (1:50, w:w) was added to the samples [17,18]. After overnight incubation at 37 °C, HDL peptide samples were acidified using 0.5% trifluoroacetic acid (Sigma-Aldrich) and desalted using the C18-StageTip protocol [19]. All steps were conducted in a single batch to eliminate inter-assay variability. Before MS analysis, samples were resuspended in 0.1% formic acid (Fisher Chemical, Zurich, Switzerland) at a concentration of 50 ng/µL. One microliter (50 ng) of each sample was injected into the LC-MS/MS system. MS proteomics data have been deposited to the Mass Spectrometry Interactive Virtual Environment (MassIVE) with access via ftp://MSV000095160@massive.ucsd.edu and doi:10.25345/C5J38KV56 (username: MSV000095160_reviewer/password: 5z6o\Z0Z2Dtu).

Liquid chromatography-mass spectrometry (LC-MS/MS) analyses

Digested HDL proteins were loaded onto a trap column (nanoViper C18, 3μm, 75μm x 2cm, Thermo Scientific, Waltham, USA) and eluted onto a C18 column (nanoViper, 2μm, 75μm x 15cm, Thermo Scientific). Peptide analyses were performed employing an Easy-nLC 1200 UHPLC system coupled to an Orbitrap Fusion Lumos tribrid mass spectrometer equipped with a nanospray FlexNG ion source (Thermo Scientific, 2150 V), in a 39 min gradient and normalized collision energy of 30 for HCD fragmentation. The LC system was initially set at a flow rate of 300 nL/min using pumps A (0.1% formic acid) and B (80% acetonitrile in 0.1% formic acid). A linear gradient from 5 to 28% B was achieved in 25 min, followed by another linear gradient from 28 to 40% B in 3 min. Solvent B was increased to 95% B in one minute to wash the system, which was maintained for another 10 min (350 nL/min), before re-equilibrating the system for another run.

Data-dependent acquisition (DDA) was used to build a spectral library. With this purpose, DDA precursor ions were identified using MS1 resolution of 120,000 (at m/z 200) with AGC target set to 5x105, maximum injection time of 50 ms, and a full scan range of 350-1550 m/z. Fragment ions were analyzed in MS2 mode with a resolution of 30000 (at m/z 200), with a standard AGC target and maximum injection time of 54 ms. For data-independent acquisition (DIA), transitions were monitored in centroid mode with a resolution of 30,000 (at m/z 200), AGC target of 5x105, precursor isolation range between 400-1000 m/z, scan range of product ions between 150-1,650 m/z, maximum injection time of 54 ms and staggered isolation windows of 24 m/z with 0.5 m/z margins. A precursor ion MS1 full scan was also acquired in profile mode between each cycle using 30,000 resolution and m/z range between 350 and 1,550. All MS analyses were performed using orbitrap as the mass analyzer and RunStart EASY-IC as internal calibration.

Sample randomization and quality controls (QCs)

DDA proteomics was used to acquire data to build a spectral library to quantify HDL proteins by DIA [20,21,18]. Thus, for DDA, 15 pools were constructed, consisting of 20 randomly selected samples.

For DIA quantifications, samples from both experimental groups were randomly injected into the MS. Twenty samples were randomly chosen to compose a pooled sample used as a quality control. This QC pooled sample was digested in 12 replicates to account for digestion variability (Digestion QC). Furthermore, a pooled QC sample was injected 19 times intercalating with samples, aiming to control for MS variability (MS QC). Finally, the 25 DDA pools were consolidated in a single aliquot and injected 12 times at the end of the batch (replicate QC ) [21].

MS data processing

MaxQuant software (v. 2.2.0.0) was used to search spectra in .raw files from DDA data against the human proteome (Uniprot, 20594 entries downloaded in February 2023). The protein detection and quantification criteria included at least two peptides (one being a unique peptide) from specific trypsin hydrolysis (up to two missed cleavages were allowed). Methionine oxidation and cysteine carbamidomethylation were selected as variable and fixed modifications, respectively. MaxQuant .msms file was converted to a .dlib file using EncyclopeDIA (v.1.12.31), employed as the library for DIA. For data processing, DIA .raw files were converted to .mzML using MSConvert (v.3.0.22297) and used within EncyclopeDIA software to quantify proteins [20]. Software configurations included the selection of 4 to 6 transitions per peptide, mass tolerance set at a resolution of 30,000, and Percolator v.3-01. Protein intensities were TIC normalized.    The output of protein intensities was used for quantification, with a specific protein being considered for quantification if coefficients of variation (CVs) lower than 25% were obtained for this protein in at least 2 out of 3 QCs (Digestion, MS or replicate QCs), after outlier removal using Grubbs’ test.. Protein intensities equal to zero were considered missing values. Keratins were considered contaminants from sample preparation and were also excluded from analysis”.

Reference [17] corresponds to Santana, Monique de Fatima Mello et al. The paper didn’t have any proteomics included.

Reference 17 has been replaced with the correct one. Thank you for the observation.

  1. Authors mentioned that Angiotensin peptide is used as the global internal standard in this study. Can you please clarify if this standard was also used for checking the instrument stability, and also used for establishing the lower limit of quantitation for each protein analyzed using PRM? 

As mentioned earlier, the correct description of the proteomic approach using DIA has been provided in detail. The authors apologize for the error.

  1. Line 146 – Authors deposited all raw mass spec files on MASSIVE. Most of the proteomics papers include other metadata files including the list of all identified proteins, peptides with retention times, and quantitative information in the form of Excel or as supplementary tables. Please include this information in the manuscript.  

The authors have uploaded to the MassIVE environment all proteomic files, including raw files (.raw), converted peak list files (.mzML), EncyclopeDIA result files (.elib), spectral library (.dlib), FASTA file (.fasta) and result files for peptides and proteins (.txt). A table with information matching sample names with experimental groups was also uploaded

  1. Line 154 – Statistical analysis – The authors mentioned that PRM data was analyzed using Skyline software. Nothing has been included in the data analysis section – on how the integration of the peptides was done and how was the data normalized etc? This is an important part of proteomic analysis. Including all details improves the paper.

 The information has been included in detail as properly suggested.

  1. The authors haven’t included the UniProt FASTA file details used for the analysis of the PRM data. Please include the date of accession of these details. Can you please include the details on how the library of specific proteins has been constructed for the analysis? How many peptides were considered, and how the selection was made, and what possible cut-offs were used to generate the library.

The authors thank the reviewer for the observation. We have included FASTA details in the methodology. We have also uploaded the UniProt FASTA file we used to the MassIVE page. In addition, we have provided detailed information regarding spectral library construction. For DIA analysis, we set the software to consider all eligible peptides. Identified proteins were filtered for quantification according to the criteria described below, also described in the methodology section: “The output of protein intensities was used for quantification, with proteins being considered for quantification if coefficients of variation (CVs) lower than 25% were obtained after outlier removal using Grubbs’ test. Protein intensities equal to zero were considered missing values. Keratins were considered contaminants from sample preparation and were also excluded from analysis.”

  1. Results: The figures shown in the paper need to include more information. It wasn’t clear what parameters were considered for the statistical differences. – are those peak areas, or fold change relative abundance or absolute quantities of the proteins? Including data analysis details as requested above would be clearer on what is considered for the statistical significance.

The information has been included as suggested  

  1. Please format citations in the discussion and keep them consistent. For example, in line no-290 – Arenas  M et al – remove M. In Line 336 – the year was missing. So, either remove the year or keep them.

The format has been as properly corrected as suggested (line 360).

  1. Authors had discussed the results of dysregulated protein expressions in urinary and serum vesicles in a paragraph (line – 384 to 388). Though this is relevant to BC, this is not adding up to the aim and hypothesis of the current paper. Please discuss this further and try to connect with the paper, or is this suggesting any future studies? Not clear.

The authors have changed the sentence making it clearer to the reader (line 448) “Proteomic analyses have been conducted on various samples, including tissues, whole plasma, and urine, to better understand the pathophysiology of BC”

  1. It would be good if authors could discuss the advantage of PRM over DIA proteomics, and how this significantly improves the understanding of HDL.

Commentaries have been included in the Discussion (line 339)” Quantification of HDL proteins is mainly obtained by data-dependent acquisition (DDA) proteomics. However, this untargeted method is stochastic and may suffer from technical variabilities. Targeted strategies enable precise quantification of low-abundance proteins with exceptional accuracy, sensitivity, and reproducibility. Targeted proteomics, represented by selected reaction monitoring (SRM) is the gold standard for MS quantification. Another targeted quantification strategy, called parallel reaction monitoring (PRM) showed comparable analytical performance to that obtained by SRM to quantify HDL proteome [17]. Data-independent acquisition (DIA), used in the present study, is a relatively recent quantification strategy in MS-based proteomics, that combines the benefits of both untargeted and targeted approaches. This method enables samples to be acquired once and analyzed multiple times in silico, as all fragment ions are recorded and assessed. Importantly, it has been demonstrated that both DIA and PRM are equally effective in differentiating HDL subclass proteomes, with comparable analytical performances and a high level of agreement [18]. Moreover, the variability of DIA for HDL proteomics was determined to be low, making this an excellent quantitative method [20].”

The proteins discussed in this study could also be identified from undepleted plasma samples – so, can you elaborate on why it is important to do HDL proteomics?

A new sentence has been added on this matter. Please see line 463 “It is important to note that HDL shares proteins with other lipoproteins in plasma, but certain proteins function differently depending on the specific lipoprotein they are associated with [61]. Moreover, some proteins, exist in plasma both as HDL-bound and in a lipid-free fraction, and they also display different properties [62]. Thus, the determination of HDL proteome adds a new piece of information to BC biology”.

  1. Conclusion section is missing.

The conclusion section has been included in the new version of the manuscript (line 468)

  1. Please check for minor editing errors, in the document. 

The entire document has been revised

Reviewer 2 Report

Comments and Suggestions for Authors

In this manuscript, the authors describe using LC-MS proteomics to compare HDL proteomes from many breast cancer types versus healthy controls.  The experiments, data analysis, and results look very good, but I have some major and minor concerns.  The subject is very important.  The manuscript is well written except that there are numerous minor grammatical errors.

Major issues:

Line 136, 138: The authors wrote that they performed “parallel reaction monitoring” (PRM) LC-MS.  They wrote that they submitted their LC-MS spectra files to MassIVE, and they included the user/password.  Their LC-MS files are the result of data independent acquisition (DIA) LC-MS, not PRM.  DIA and PRM are very different.  The LC-MS instrument method(s) need to be described accurately and in more detail in the Methods section.  The authors should describe: the LC mobile phases and gradient, the ESI voltage, the MS1 scan range, the MS1 resolution setting, the MS2 resolution setting, the MS2 precursor isolation window width, and the collision energy.

The authors need to describe the software they used to analyze the LC-MS spectra, and what settings they used.  They need to describe the protein sequence database FASTA they used, and if they performed a spectral library-free DIA analysis, or if they used a spectral library, and if so, they need to describe the spectral library that they used.  They need to say if they included a dataset of LC-MS common contaminants in their analysis.  They need to say what peptide-spectrum-match and protein identification false discovery rate requirement they used.

The peptide and protein identification and quantitation result file(s) and FASTA file(s) were not included in their submission to MassIVE.  They need to be included so that they are available to the public.  A detailed table that describes and links of all of the experimental groups, samples, the raw LC-MS files, and the DIA results file(s) needs to be included with the submission with sufficient detail such that people could properly reanalyze the data.

The order that the samples were run on the LC-MS needs to be described.  These samples were stable isotope label free, and apparently no global normalization was performed.  Therefore, this analysis was very sensitive to LC-MS instrument performance.  Ideally, the samples would have been blocked and randomized.

Minor Issues:

Figures 1, 3, 4, 6: The y-axes are doubly labeled with 10^N (e.g., 10^8), and this should only appear once.  Also, the measurement type needs to be described (e.g., the sum of the MS signal intensity).

Figures 2, 5, 7: The value of the color bar needs to be described (e.g., z-score).

A supplemental table of the LC-PRM data should be included with the article.  The rows should be the proteins, and the columns should be the samples, and the values should be the abundance values.  The breast cancer type and stage should be indicated for each sample.

Line 240: “Figure 4” should be “Figure 5”.

Line 260: “Figure 6” should be “Figure 7”.

Comments on the Quality of English Language

Minor issues.

Author Response

Reviewer #2

In this manuscript, the authors describe using LC-MS proteomics to compare HDL proteomes from many breast cancer types versus healthy controls.  The experiments, data analysis, and results look very good, but I have some major and minor concerns.  The subject is very important.  The manuscript is well written except that there are numerous minor grammatical errors.

The authors thank the reviewer for the insightful comments and valuable suggestions, which have significantly improved the quality of the manuscript. Below is a detailed response to each comment. Revisions are highlighted in red in the latest version of the manuscript.

Major issues:

Line 136, 138: The authors wrote that they performed “parallel reaction monitoring” (PRM) LC-MS.  They wrote that they submitted their LC-MS spectra files to MassIVE, and they included the user/password.  Their LC-MS files are the result of data independent acquisition (DIA) LC-MS, not PRM.  DIA and PRM are very different.  The LC-MS instrument method(s) need to be described accurately and in more detail in the Methods section.  The authors should describe: the LC mobile phases and gradient, the ESI voltage, the MS1 scan range, the MS1 resolution setting, the MS2 resolution setting, the MS2 precursor isolation window width, and the collision energy.   

The authors need to describe the software they used to analyze the LC-MS spectra, and what settings they used.  They need to describe the protein sequence database FASTA they used, and if they performed a spectral library-free DIA analysis, or if they used a spectral library, and if so, they need to describe the spectral library that they used.  They need to say if they included a dataset of LC-MS common contaminants in their analysis.  They need to say what peptide-spectrum-match and protein identification false discovery rate requirement they used.

The peptide and protein identification and quantitation result file(s) and FASTA file(s) were not included in their submission to MassIVE.  They need to be included so that they are available to the public.  A detailed table that describes and links of all of the experimental groups, samples, the raw LC-MS files, and the DIA results file(s) needs to be included with the submission with sufficient detail such that people could properly reanalyze the data.

 The order that the samples were run on the LC-MS needs to be described.  These samples were stable isotope label free, and apparently no global normalization was performed.  Therefore, this analysis was very sensitive to LC-MS instrument performance.  Ideally, the samples would have been blocked and randomized.

The authors apologize for the error, as the methodology described was based on our previous publication on HDL proteomic analysis by PRM. A solution digestion was utilized. In the new version of the manuscript, the method used in the present investigation for HDL proteomic assessment by DIA is described in detail including the information required by the reviewer (lines 138 to 201).

“HDL digestion for proteomics

The Bradford assay (Bio-Rad, Hercules, USA) determined total protein concentration in isolated HDL. Five micrograms of HDL-protein were diluted in 50 mM ammonium bicarbonate buffer (Sigma-Aldrich, St. Luis, USA) containing 0.01% ProteaseMAX MS-compatible surfactant (Promega). Proteins were reduced with 5 mM dithiothreitol (Bio-Rad), for 1h at 37 °C, alkylated with 15 mM iodoacetamide (Bio-Rad) for 30 min and excess iodoacetamide was quenched using 2.5 mM dithiothreitol for 15 min at room temperature RT. Proteins were then digested with trypsin [1:40, w:w, trypsin (Promega, Madison, USA): HDL proteins] for 4h at 37 °C. A second aliquot of trypsin (1:50, w:w) was added to the samples [17,18]. After overnight incubation at 37 °C, HDL peptide samples were acidified using 0.5% trifluoroacetic acid (Sigma-Aldrich) and desalted using the C18-StageTip protocol [19]. All steps were conducted in a single batch to eliminate inter-assay variability. Before MS analysis, samples were resuspended in 0.1% formic acid (Fisher Chemical, Zurich, Switzerland) at a concentration of 50 ng/µL. One microliter (50 ng) of each sample was injected into the LC-MS/MS system. MS proteomics data have been deposited to the Mass Spectrometry Interactive Virtual Environment (MassIVE) with access via ftp://MSV000095160@massive.ucsd.edu and doi:10.25345/C5J38KV56 (username: MSV000095160_reviewer/password: 5z6o\Z0Z2Dtu).

Liquid chromatography-mass spectrometry (LC-MS/MS) analyses

Digested HDL proteins were loaded onto a trap column (nanoViper C18, 3μm, 75μm x 2cm, Thermo Scientific, Waltham, USA) and eluted onto a C18 column (nanoViper, 2μm, 75μm x 15cm, Thermo Scientific). Peptide analyses were performed employing an Easy-nLC 1200 UHPLC system coupled to an Orbitrap Fusion Lumos tribrid mass spectrometer equipped with a nanospray FlexNG ion source (Thermo Scientific, 2150 V), in a 39 min gradient and normalized collision energy of 30 for HCD fragmentation. The LC system was initially set at a flow rate of 300 nL/min using pumps A (0.1% formic acid) and B (80% acetonitrile in 0.1% formic acid). A linear gradient from 5 to 28% B was achieved in 25 min, followed by another linear gradient from 28 to 40% B in 3 min. Solvent B was increased to 95% B in one minute to wash the system, which was maintained for another 10 min (350 nL/min), before re-equilibrating the system for another run.

Data-dependent acquisition (DDA) was used to build a spectral library. With this purpose, DDA precursor ions were identified using MS1 resolution of 120,000 (at m/z 200) with AGC target set to 5x105, maximum injection time of 50 ms, and a full scan range of 350-1550 m/z. Fragment ions were analyzed in MS2 mode with a resolution of 30000 (at m/z 200), with a standard AGC target and maximum injection time of 54 ms. For data-independent acquisition (DIA), transitions were monitored in centroid mode with a resolution of 30,000 (at m/z 200), AGC target of 5x105, precursor isolation range between 400-1000 m/z, scan range of product ions between 150-1,650 m/z, maximum injection time of 54 ms and staggered isolation windows of 24 m/z with 0.5 m/z margins. A precursor ion MS1 full scan was also acquired in profile mode between each cycle using 30,000 resolution and m/z range between 350 and 1,550. All MS analyses were performed using orbitrap as the mass analyzer and RunStart EASY-IC as internal calibration.

Sample randomization and quality controls (QCs)

DDA proteomics was used to acquire data to build a spectral library to quantify HDL proteins by DIA [20,21,18]. Thus, for DDA, 15 pools were constructed, consisting of 20 randomly selected samples.

For DIA quantifications, samples from both experimental groups were randomly injected into the MS. Twenty samples were randomly chosen to compose a pooled sample used as a quality control. This QC pooled sample was digested in 12 replicates to account for digestion variability (Digestion QC). Furthermore, a pooled QC sample was injected 19 times intercalating with samples, aiming to control for MS variability (MS QC). Finally, the 25 DDA pools were consolidated in a single aliquot and injected 12 times at the end of the batch (replicate QC ) [21].

MS data processing

MaxQuant software (v. 2.2.0.0) was used to search spectra in .raw files from DDA data against the human proteome (Uniprot, 20594 entries downloaded in February 2023). The protein detection and quantification criteria included at least two peptides (one being a unique peptide) from specific trypsin hydrolysis (up to two missed cleavages were allowed). Methionine oxidation and cysteine carbamidomethylation were selected as variable and fixed modifications, respectively. MaxQuant .msms file was converted to a .dlib file using EncyclopeDIA (v.1.12.31), employed as the library for DIA. For data processing, DIA .raw files were converted to .mzML using MSConvert (v.3.0.22297) and used within EncyclopeDIA software to quantify proteins [20]. Software configurations included the selection of 4 to 6 transitions per peptide, mass tolerance set at a resolution of 30,000, and Percolator v.3-01. Protein intensities were TIC normalized.    The output of protein intensities was used for quantification, with a specific protein being considered for quantification if coefficients of variation (CVs) lower than 25% were obtained for this protein in at least 2 out of 3 QCs (Digestion, MS or replicate QCs), after outlier removal using Grubbs’ test. Protein intensities equal to zero were considered missing values. Keratins were considered contaminants from sample preparation and were also excluded from analysis”.

The authors have uploaded to the MassIVE environment all proteomic files, including raw files (.raw), converted peak list files (.mzML), EncyclopeDIA result files (.elib), spectral library (.dlib), FASTA file (.fasta) and result files for peptides and proteins (.txt). A table with information matching sample names with experimental groups was also uploaded

Minor Issues:

Figures 1, 3, 4, 6: The y-axes are doubly labeled with 10^N (e.g., 10^8), and this should only appear once.  Also, the measurement type needs to be described (e.g., the sum of the MS signal intensity). 

The figures have been corrected.

Figures 2, 5, 7: The value of the color bar needs to be described (e.g., z-score). 

The authors have included a description

A supplemental table of the LC-PRM data should be included with the article.  The rows should be the proteins, and the columns should be the samples, and the values should be the abundance values.  The breast cancer type and stage should be indicated for each sample. 

The authors agree and all data have been included in the repository as stated in line 151 “MS proteomics data have been deposited to the Mass Spectrometry Interactive Virtual Environment (MassIVE) with access via ftp://MSV000095160@massive.ucsd.edu and doi:10.25345/C5J38KV56 (username: MSV000095160_reviewer/password: 5z6o\Z0Z2Dtu)”, and line 245 “Tables with protein intensities and information regarding samples have been included in the MassIVE repository”.

Line 240: “Figure 4” should be “Figure 5”.

 The number of the figure in the text was corrected as properly suggested (line 295)

Line 260: “Figure 6” should be “Figure 7”.

 The number of the figure in the text was corrected as properly suggested (line 316)

Round 2

Reviewer 2 Report

Comments and Suggestions for Authors

All of my concerns have been fully addressed, and I now recommend that the manuscript be accepted for publication.

Comments on the Quality of English Language

Only minor grammatical issues.